# Reducing Atomic Clashes in Geometric Diffusion Models for 3D Structure-Based Drug Design

## Abstract

In the domain of Three-dimensional Structure-Based Drug Design (3D SBDD), the 3D spatial structures of target pockets serve as inputs for the generation of molecular geometric graphs. The Geometric Diffusion Model (GDM) has been recognized as the state-of-the-art (SOTA) method in 3D SBDD, attributed to its exceptional generation capabilities on geometric graphs. However, the inherent data-driven nature of GDM occasionally neglects critical inter-molecular interactions, such as Van der Waals force and Hydrogen Bonding. Such omissions could produce molecules that violate established physical principles. Particular evidence is that GDMs exhibit atomic clashes during generation due to the overly close proximity of generated molecules to protein structures. To address this, our paper introduces a novel constrained sampling process designed to obviate such undesirable collisions. By integrating a non-convex constraint within the current Langevin Dynamics (LD) of GDM and utilizing the proximal regularization techniques, we force molecular coordinates to obey the imposed physical constraints. Notably, the proposed method requires no modifications to the training process of GDMs. Empirical evaluations show a significant reduction in atomic clashes via the proposed method compared to the original LD process of GDMs.

## 1 Introduction

Structure-based drug design plays a crucial role in the drug development process because it enables scientists to predict how different drugs will interact with their target proteins at the molecular level (Anderson, 2003). It utilizes the knowledge of the three-dimensional (3D) structures of biological targets, allowing for the generation and optimization of compounds with high affinity and specificity. With the recent advancements in deep learning, researchers are continuously exploring how to use deep learning to design structure-based generative models. Unlike models that generate small molecules from scratch (Luo & Ji, 2022), without considering protein information, structure-based generative models integrate the intricate details of protein structures to create more informed and contextually relevant molecules. Clearly, such models have the potential to greatly accelerate the discovery of new drugs against known specific targets (Dara et al., 2022).

The incorporation of 3D structure information significantly enhances the process of drug design (Adcock & McCammon, 2006; Hollingsworth & Dror, 2018; Yuriev & Ramsland, 2013). There is a prevalent set of approaches that incorporate 3D structure information into the autoregressive flow model (Rezende & Mohamed, 2015; Dinh et al., 2014; Gebauer et al., 2019). For instance, The GraphBP (Liu et al., 2022) and the cG-SchNet (Gebauer et al., 2022), both of which are autoregressive flow models, employ an atom-by-atom method to generate molecules designed to bind to specific proteins. However, this atom-by-atom method may not always be optimal when generating molecules based on the pocket, as it has the potential to produce chemically invalid intermediates.

Due to the limitations associated with the autoregressive model, Hoogeboom et al. (2022) pioneered the use of the diffusion model for the task of molecule generation. This model holds an advantage as it generates molecules in their entirety all at once, avoiding the creation of invalid chemical intermediates like the autoregressive models. Building on this success, several other structure-based molecular diffusion models such as Targetdiff (Guan et al., 2023a), DiffSBDD (Schneuing et al.,

2022), and PMDM (Huang, 2023) have been proposed. Each of these models possesses a distinctive design. In Targetdiff, uniform noise is added to the atom types. DiffSBDD introduces an inpainting method to generate molecules during inference time. PMDM, on the other hand, employs a dual equivariant score kernel network based on the distance between atoms. According to experimental results, all these models have demonstrated the capability to generate molecules with lower docking scores (indicating higher affinity) compared to reference molecules. However, these methods predominantly rely on data-driven approaches to simulate the spatial relationship between molecules and proteins, thereby overlooking certain physical rules.

To illustrate this point, for instance, in the sample results of Targetdiff, a phenomenon that we defined as "atomic clash" is observed, indicating that some generated molecules maintain an inappropriate spatial relationship with the corresponding pocket. This observation is grounded on the assumption (Hooft et al., 1996) that atoms being overly proximate in protein structure constitute an error, coupled with the common bond length (Lide et al., 2005). We postulate that any distance less than 2Å between the atoms of generated molecules and the amino acid atoms is incorrect and could potentially be construed as a bond (See Figure 1.a). We conducted an analysis of the distribution of the atomic gap between the ligand and protein within the CrossDock2020 dataset. The results of the statistical analysis reveal that gap distances of less than 2Å are virtually non-existent in the database (See Figure 1.b). We also provide the statistic results of the PDBbind dataset (see Appendix C.1). Additionally, Harris et al. (2023) believe that the non-clash distance is based on the bond length plus a 0.5Å threshold, which is closely aligned with our definition. For instance, the typical C-C bond length is around 1.5Å, and adding 0.5Å results in 2.0Å. To address the atomic clash issue, Guan et al. (2023b) attempts to refine the sampling process by utilizing the gradient of clash constraints, but the direct application of the gradient algorithm may potentially induce errors, especially when dealing with non-smooth constraints.

In this paper, we formulate the issue of the spatial clash between the molecule and protein as a constrained optimization problem, conceptualizing it based on the observed "atomic clash" phenomena in models like Targetdiff. We enhance the original sampling process based on Langevin Algorithm (LA) by integrating a proximal operator to ascertain the optimal solution under such constraints. Concurrently, in comparison to the previously mentioned method, we conducted a theoretical exploration to elucidate the reasons behind its incremental improvement. The key contributions of our study are outlined as follows:

- We employ the proximal operator to tackle the non-smooth distance constraints, subsequently diminishing the frequency of atomic clashes.
- Our methods avoid the necessity for any training process specific to the structure-based diffusion model.
- We conduct a theoretical validation to confirm the convergence of our approach.

In Section 3, we present the foundational knowledge necessary for understanding our research. In Section 4, we provide a detailed explanation of our proposed method. In Section 5, we first compare our method with existing models, wherein the comparative results demonstrate that our approach achieves a lower clash ratio. Subsequently, we investigate the influence of the parameters of our method on the final outcome through an ablation study. Finally, we evaluate the impact of our method on the chemical properties of the generated molecules.

## 2 RELATED WORK

**Molecule Generation** The advancement of the generative model has led to the application of various models such as Variational Autoencoders (VAE), Generative Adversarial Networks (GAN), Flow, and Denoising Diffusion Probabilistic Models (DDPM) in diverse molecule generation methods. Xu et al. (2021) proposed a SMILES generation model based on the conditional RNN (Lipton et al., 2015) framework, wherein protein information was treated as conditional input. In a different approach, the DeepTarget model (Chen et al., 2023) generates SMILES of molecules from protein target sequences. One notable approach is the JT-VAE (Jin et al., 2018), which generates molecules fragment-by-fragment. This process involves two phases: the generation of coarse and valid chemical substructure, followed by assembling these fragments into a complete molecule. In contrast, the GF-VAE (Ma & Zhang, 2021) and Two-step Graph VAE (Bresson & Laurent, 2019) generate

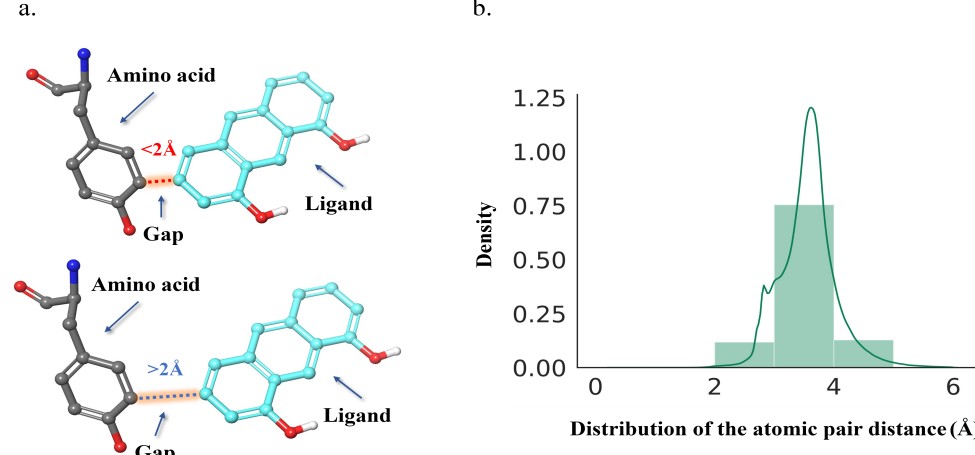

Figure 1: a. Two distance situations between amino acids and molecules are depicted: the upper half represents an inappropriate spatial distance ($< 2$Å), while the lower half portrays an appropriate spatial distance ($> 2$Å). b. Statistical distribution of pairwise distances between proteins and molecules within 6Åin the Crossdocked2020 Dataset.

molecules by incrementally adding atoms and bonds. Training GAN models for molecule generation is challenging compared to VAE models. One attempt of GAN models calls LatentGAN (Prykhodko et al., 2019) which is a molecule generation model that generates molecules in SIMILES notation. Building upon this, Bai et al. (2021) proposed MolGAN, a GAN-based architecture capable of generating more meaningful drug-like molecules. MoFlow (Zang & Wang, 2020), GraphAF (Shi et al., 2020), GraphBP (Liu et al., 2022), and GraphDF (Luo et al., 2021b) take a different approach by generating molecules through the sequential addition of bonds and then atoms, contrasting with the aforementioned VAE models' methods. Furthermore, both MDM (Huang et al., 2023), MolDiff (Peng et al., 2023) and MiDi (Vignac et al., 2023) are end-to-end molecular generation models that have been trained using analogous diffusion frameworks. Specifically, MolDiff and MiDi construct bonds directly from the trained model through a bond diffusion process, unlike MDM.

**Structure-based Drug Design** The primary objective of structure-based drug design is to identify potential molecules that can specifically target a protein, thereby inducing effective conformation changes in the protein. In a recent study, Kang et al. (2022) utilized a conditional variational autoencoder model (VAE) for the generation of molecular graphs. However, Their model did not account for the inherent spatial relationship within the ligand-protein complex. In contrast, Ragoza et al. (2022) proposed a cVAE model trained on an atomic density grid representation of a protein-ligand complex. They utilized an atom fitting algorithm and a bond inference procedure to transform the atomic density grids into a discrete 3D molecular structure. Building on the assumption that a drug's shape undergoes minimal changes upon binding to a pocket, Long et al. (2022) developed a model that generates molecules conditioned on sketching reasonable shapes from protein pockets. This innovative approach provides a new perspective on drug design. Furthermore, considering the interaction between the functional groups of molecules and protein residues, the FLAG model (Zhang et al., 2022) can generate the molecules motif-by-motif in a 3D perspective. These methods offer a more detailed and comprehensive view of molecule generation based on the pocket.

## 3 PRELIMILARIES

### 3.1 SCORE MATCHING MODEL

Directly modeling the data distribution $p_{\text{data}}(x)$ using a neural network $p_\theta(x)$ is challenging (Song & Ermon, 2019). To avoid this, they learn the gradients of the perturbed data distribution and obtain

samples via Langevin dynamics. From an energy perspective, the $p_\theta(x)$ can be expressed as

$$p_\theta(x) = \frac{1}{Z_\theta} e^{-U(x)}. \tag{1}$$

where $U(x) := f_\theta(x)$, an arbitrarily flexible and parameterizable energy function often modeled by a neural network. $Z_\theta$ serves as a normalizing constant to guarantee that $\int p_\theta(x)dx = 1$. Computing the normalizing constant is intractable, so both the frameworks of the denoising score matching model and the denoising diffusion probabilistic model aim to learn a network $s_\theta(\tilde{x}, t)$ that accurately represents the gradient of the perturbed data distribution $\nabla_{\tilde{x}} \log p(\tilde{x}|x)$ (Song et al., 2020).

**Perturbing Data** As same to the diffusion model, the aim of the score-matching model is to establish a diffusion process $\{x_t\}_{t=0}^T$, which is indexed by a continuous time variable $t \in [0, T]$. This process ensures that $x_0$ and $x_T$ correspond to the data distribution, $v$, and a tractable form (prior distribution) for efficiently generating samples, respectively. The diffusion process can be modeled by the following SDE:

$$d\mathbf{x} = \mathbf{f}(x, t)dt + \eta(t)d\mathbf{w}, \tag{2}$$

where $\mathbf{w}$ symbolizes the Brownian motion, $\mathbf{f}(\cdot, t)$ denotes the drift coefficient of $x_t$, and $\eta(\cdot)$ is the diffusion coefficient of $x_t$. Let $v_t := \mathrm{Law}(x_t)$ be the measures along the process above. The forward process can be interpreted as a transformation of samples from the data distribution $v$ into noises.

**Reverse SDEs** Interestingly, Anderson (1982) posits that the reverse of a diffusion process also constitutes a diffusion process, which generates samples by running backward in time and is characterized by the reverse-time Stochastic Differential Equation (SDE)::

$$d\mathbf{x} = [\mathbf{f}(x, t) - \eta(t)^2 \nabla_\mathbf{x} \log p_t(x)]dt + \eta(t)d\bar{\mathbf{w}}, \tag{3}$$

where $\bar{\mathbf{w}}$ represents the Brownian motion when time flows backward from $T$ to $0$. Notably, from Eq.3, if we can obtain $\nabla_\mathbf{x} \log p_t(x)$ for all $t$, we can derive the reverse diffusion process and simulate it to acquire $p_0$ from $p_T$.

**Traing and Sampling** According to the Eq.3, a score neural network $s_\theta$ is required to model the $\nabla_\mathbf{x} \log p_t(x)$. A direct and efficient approach is to calculate the Mean Squared Error (MSE) loss between them, as previously mentioned. The continuous generalization can be expressed as:

$$\theta^* = \underset{\theta}{\mathrm{argmin}} \, \mathbb{E}_t \{ \varphi(t) \mathbb{E}_{x_0} \mathbb{E}_{x_t|x_0} \left[ \|s_\theta(x_t, t) - \nabla_{x_t} \log p(x_t|x_0)\|_2^2 \right] \}, \tag{4}$$

where $\theta^*$ is the optimal parameter of the trained model. The $\varphi(t)$ is a weight function depending on the time step $t$. Given sufficient data and model capacity, the score-matching is trained by Eq.3, denoted as $s_{\theta^*}(x, t)$, which approximates $\nabla_\mathbf{x} \log p_t(x)$ for nearly all $x$ and $t$. For sampling purposes, once the trained score-matching network is obtained, the standard Langevin Dynamics Sampling method (Song & Ermon, 2019; Pierzchlewicz, 2022) can be employed to generate the target sample, which can be sequentially executed in preset steps using this sample equation:

$$x_{t-1} = x_t + \epsilon_t \nabla_\mathbf{x} \log p_t(x) + \sqrt{2\epsilon_t}\mathbf{z}, \quad t = 1, 2, \ldots, T, \tag{5}$$

where $z \sim \mathcal{N}(0, 1)$ and $\epsilon_t$ represents a step size, which serves as a hyper-parameter depending on time $T$. After iterating T times, the final sample $x_T$ is obtained from its distribution $p(x_T)$, and this distribution is approximately equal to the data distribution $p_{data}(x)$.

### 3.2 PROXIMAL REGULARIZATION

For the task of unconstrained minimization of a continuously differentiable function (denoted $f(x)$), the gradient descent algorithm is often used. This method seeks to minimize the function $f(x)$ by iteratively applying:

$$x_k = x_{k-1} - t_k \nabla f(x_{k-1}), \quad x_0 \in \mathbb{R}, \tag{6}$$

Considering the regularized problem, a function $g(x)$ is incorporated into the equation 6. As a result, the general formulation can be expressed as $\min F(x) \equiv f(x) + g(x) : x \in \mathbb{R}^n$. Given a point $x_{k-1}$, a quadratic approximation of $F(x)$ (Beck & Teboulle, 2009) can be obtained:

$$\underset{x}{\mathrm{argmin}} := f(x_{k-1}) + \langle x_k - x_{k-1}, \nabla f(x_{k-1}) \rangle + \frac{1}{2\lambda} \|x_k - x_{k-1}\|^2 + g(x_k). \tag{7}$$

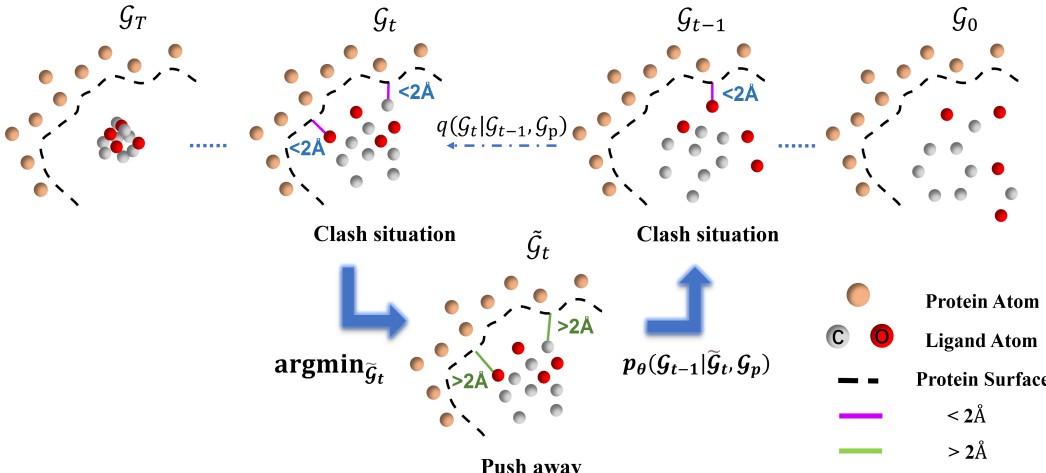

Figure 2: Simplified schematic of the sampling process with the incorporation of the constraint formula. The dotted line denotes the surface of the protein pocket, while $\tilde{\mathcal{G}}_t$ denotes the information of the generated molecule after incorporating constraints. $\mathcal{G}_t$ represents the information on molecules normally generated by the model.

By disregarding constant terms in $x_{k-1}$:

$$\text{prox}_g(y) = \underset{x}{\text{argmin}}\{g(x) + \frac{1}{2\lambda}\|x - y\|^2\}. \tag{8}$$

where the $\frac{1}{2\lambda}$ plays the role of a step size. Clearly, in accordance with the proximal mapping Eq.8, the value of $x_k$ can be computed via $\text{prox}_g(x_{k-1})$.

## 4 METHOD

**Notation** Let $\mathcal{G}_l = (x_l, v_l)$ denote the 3D generated molecular geometric structure, where $x_l$ represents the atomic positions and $v_l$ encompasses the atomic features (e.g. atom type, atom charge). Similarly, we employ $\mathcal{G}_p = (x_p, v_p)$ to represent the geometric structure of the protein pocket. Here, $x_p$ and $v_p$ correspond to the positions and types of protein atoms, respectively.

### 4.1 ENERGY CONSTRAINT

Recalling Eq.1 and considering the position $x_l$ of molecules, we introduce a constraint function $g(x_l)$ as:

$$\min_{x_l} \quad U(x_l) := f_\theta(x_l) \tag{9}$$

$$\text{s.t.} \quad g(x_l) \leq 0. \tag{10}$$

In order to tackle the atomic clash problem, we define $g(x_l)$ as a function that return larger values when the distance between the $x_l$ and $x_p$ is too close. Additionally, when the distance between them exceeds 2Å, $g(x_l)$ should no longer impose a constraining effect. Therefore, to satisfy the above conditions, we design $g(x_l)$ as a piece-wise function:

$$g(x_l) = \begin{cases} \sum_{k=1}^{K} e^{-(x_l - x_p^k)^2/\sigma}, & \text{if } d \leq 2 \\ 0, & \text{if } d > 2, \end{cases}$$

where $d = min(norm(x_l - x_p))$ refers to the closed paired distance of the molecule's and protein's atoms. $k$ represents the $k$ nearest protein atoms surrounding the molecular atom. If $g(x_l)$ satisfies the constraint in Eq. 10, it indicates that all distances between the atoms in $x_l$ and $x_p$ are larger than 2Å.

Then, we utilize the Lagrange Multiplier to incorporate the constraint $g(x_l)$ into the optimization object:

$$U(x_l) := f_\theta(x_l) + g(x_l). \tag{11}$$

According Eq.5 and Eq.10, we can rewrite the Eq.5 as:

$$x_l^{t-1} = x_l^t - \epsilon_t \nabla_{\mathbf{x}}(f_\theta(x_l^t) + g(x_l^t)) + \sqrt{2\epsilon_t}\mathbf{z}, t = 1, 2, \ldots, T. \tag{12}$$

Here, $-\nabla_{\mathbf{x_1}}(f_\theta(x_l)) \approx s_\theta(x_l)$ is approximated by the trained diffusion model. However, the designed $g(x_l)$ is a piece-wise function, which is non-smooth and non-convex, making the computation of $\nabla g(x_l)$ challenging. Consequently, to avoid this, we adapt the $\text{prox}_g$ to determine the optimal solution:

$$\text{prox}_g(x_l^t) = \underset{\tilde{x}_l^t}{\operatorname{argmin}}\{g(\tilde{x}_l^t) + \frac{1}{2\lambda}\|\tilde{x}_l^t - x_l^t\|^2\}, \tag{13}$$

where $x_l^t$ is derived from the Eq.12, $\lambda$ determines the impact of the proximal terms on the gradient of this equation during optimization. Alternatively, we find that the proximal formula can be interpreted as two parts: one conceptualized as **protein coordinates constrained**which evaluates the relationship between the updated molecule positions and the protein positions. This effectively directs the ligand towards a more optimal position and mitigates excessive attraction to the protein atoms; the other is conceptualized as **molecular coordinates constrained**, which is employed to penalize significant deviations in the molecule's position during optimization. By maintaining the proximity of $\tilde{x}_l^t$ to $x_l^t$, this term ensures the molecule remains relatively consistent with its preceding position. Figure 2 visually illustrates the significant impact of our constrained formula in addressing clash issues encountered throughout the Langevin Dynamic (LD) sampling procedure.

### 4.2 PROXIMAL SAMPLING

At each sample time step $t$, we adjust the position of each atom within the molecule to ensure that they maintain an adequate distance from the protein surface. This modified structure is then utilized as the input for the structure-based diffusion model to generate the subsequent sample at time step $t - 1$. Regardless of whether the distance of the next step may still be improper (less than 2Å) or not, our constrained formula consistently remains effective in rectifying such situations. To implement the process described above, and drawing inspiration from the Stochastic Proximal Langevin Algorithm (Salim et al., 2019), we divide the solution process (sampling procedure) into three distinct stages, delineated as follows:

$$h_l^{t-1} = x_l^t + \epsilon_t s_\theta(\mathcal{G}_t, \mathcal{G}_p, t)$$
$$x_h^{t-1} = h_l^{t-1} + \sqrt{2\epsilon_t}z_l^{t-1}$$
$$\tilde{x}_l^{t-1} = \text{prox}_g(x_h^{t-1})$$
$$x_l^{t-1} = \tilde{x}_l^{t-1},$$

where $\text{prox}_g$ is our designed proximal function, $z_l^{t-1}$ represents standard Gaussian random variables. Following the above sequence of equations, we initially obtain the $h_l^{t-1}$ and subsequently incorporate noise to derive $x_h^{t-1}$. Ultimately, we address the proximal function to acquire $\tilde{x}_l^{t-1}$, which corresponds to the final $x_l^{t-1}$. The algorithm details of employing our constrained formula are summarized in Appendix A.

### 4.3 CONVERGENCE ANALYSIS

We consider our sampling process as a novel Proximal Inexact Langevin Algorithm (PILA), where if the target distribution $\upsilon := p_\theta(x)$ satisfies $\alpha$-LSI and then along the Langevin dynamics (Eq. 2), KL divergence is decreasing exponentially fast. As they proved (Yingxi Yang & Wibisono, 2022) when $s$ is an approximation of score function $s_\upsilon$ and it has bounded MGF error, We show similar convergence rates accompanied by an additional bias term caused by the score estimation error.

**Theorem 1** (Convergence of KL divergence for PILA). *Assume $\upsilon$ is $\alpha$-LSI, $f_\theta$ is L-smooth, $\frac{1}{2\lambda}$ is the proximal step size, and score estimator $s$ is $L_s$-Lipschitz and has bounded MGF error ($\epsilon_{mgf}$ assumption) with $r = \frac{9}{\alpha}$. if $0 < h < \min(\frac{\alpha}{12(L_s+\lambda^{-1})(L+\lambda^{-1})}, \frac{1}{2\alpha})$, then after $k$ iterations of PILA (Eq. 12):*

$$H_\upsilon(\rho_k) \le e^{-\frac{1}{4}\alpha hk}H_\upsilon(\rho_0) + C_1 dh + C_2 \epsilon_{mgf}^2$$

*where $C_1 = 128(L_s + \lambda^{-1})(L_s + L + 2\lambda^{-1})/\alpha$ and $C_2 = 8/(3\alpha)$.*

We provide the proof of Theorem 1 as well as related notations and assumptions in Appendix B.

## 5 EXPERIMENT

In this section, we thoroughly assess our proposed approach by examining it from three distinct perspectives. Firstly, we apply our constrained formula to the TargetDiff (Guan et al., 2023a) model and subsequently compare its performance with three alternative models. The experimental results demonstrate that our approach achieves a lower mean clash ratio. Secondly, we conduct ablation studies to investigate the impact of our method on the overall performance. Lastly, we evaluate the chemical properties of the samples generated by our method, with the findings suggesting that our approach does not affect the original performance.

### 5.1 SETUP

**Data** All models were trained on the CrossDocked2020 (Francoeur et al., 2020) dataset. and their overall data preprocessing approach closely aligns with the methodology employed by Luo et al. (2021a). For convenience, we utilize a pre-divided test dataset for validation purposes. The pre-divided test dataset consists of 100 test protein samples.

**Evaluation** Following previous works (Hoogeboom et al., 2022), we employ a selection of metrics to effectively evaluate the influence of our proposed approach on the generated sampling outcomes.: (1) **Validity** refers to the proportion of generated molecules that adhere to the valence rules of the RDkit (Bento et al., 2020); (2) **Novelty** measures the proportion of generated molecules specific to a protein that are dissimilar to the reference ligand; (3) **Uniqueness** is the percentage of the unique and connected molecule among all generated molecules; (4) **Diversity** assesses the variety of generated molecules for a specific pocket. In addition, we construct three **new metrics** to evaluate our method comprehensively: (1) **Connectivity** denotes the percentage of all generated molecules that do not contain any fragment; (2) **Mean Clash Ratio** calculates the percentage of clash molecules in all generated molecules. A clash is detected when at least one pair of atomic distances between the molecule's and protein's atoms is less than the 2Å threshold.; (3) **Stability** represents the proportion of connected and not-clashing molecules in all the generated molecules, which can be calculated by $Connectivity * (1 - MeanClashRatio)$.

Table 1: Displays the performance of two types of models based on sampling evaluation criteria: PMDM and DiffSBDD, which do not account for the clash situation, and Decompdiff and Targetdiff models, which consider the clash situation. ↑: represents that a larger value signifies better performance. ↓: indicates that a smaller value corresponds to superior performance.

| Model | Validity ↑ | Novelty ↑ | Uniqueness ↑ | Diversity ↑ | Connectivity ↑ | Mean CR ↓ | Stability ↑ |
|---|---|---|---|---|---|---|---|
| PMDM | 100% | 100% | 99.31% | 0.7505 | 82.13% | 43.44% | 46.45% |
| DiffSBDD | 100% | 100% | 97.44% | 0.8246 | 54.78% | 17.17% | 45.37% |
| Decompdiff | 97.33% | 100% | 93.89% | 0.7984 | 76.13% | 9.87% | 68.61% |
| Targetdiff(ours) | 98.31% | 100% | 98.77% | 0.7107 | **84.31%** | **4.38%** | **80.62%** |

### 5.2 MODEL CLASH RESULT COMPARISON

We evaluated each model using the test set containing an equal number of proteins for every model. For each protein, we generated one hundred molecules separately. Subsequently, we calculated the clash probability between the one hundred molecules and their corresponding proteins. Finally, we selected the proteins with a probability greater than 5% in each model to construct their respective clash data sets. As shown in Table 1, the PMDM (Huang, 2023) model and the DiffSBDD (Schneuing et al., 2022) model do not consider potential clash problem between molecules and proteins during the molecule generation process, resulting in a relatively high overall collision probability.

Table 2: Compare the sampling results of the constrained formula with the original sampling results without constraints under different constraint times, different k values, and different coefficients.

| Time | Influence Factor | Validity ↑ | Diversity ↑ | Novelty ↑ | Uniqueness ↑ | Connectivity ↑ | Mean CR ↓ | Stability ↑ |
|------|------------------|-----------|-------------|-----------|--------------|----------------|-----------|-------------|
| | No constrained | 98.31% | 0.7107 | 100% | 98.77% | 85.38% | 12.31% | 74.86% |
| 150-0 | $K=1, \lambda=10$ | 98.15% | 0.7117 | 100% | 98.69% | 84.31% | 4.38% | **80.62%** |
| | $K=3, \lambda=10$ | 98.15% | 0.7109 | 100% | 98.77% | 85.38% | 7.08% | 79.34% |
| | $K=5, \lambda=10$ | 98.15% | 0.7102 | 100% | 98.77% | 85.38% | 9.00% | 77.70% |
| 300-0 | $K=1, \lambda=10$ | 97.69% | 0.7109 | 100% | 98.69% | 82.15% | 4.31% | 78.61% |
| | $K=3, \lambda=10$ | 98.31% | 0.7111 | 100% | 98.77% | 84.92% | 7.69% | 78.39% |
| | $K=5, \lambda=10$ | 98.31% | 0.7112 | 100% | 98.77% | **86.00%** | 8.00% | 79.12% |
| 500-0 | $K=1, \lambda=10$ | 98.15% | 0.7062 | 100% | 98.77% | 79.38% | **3.23%** | 76.82% |
| | $K=3, \lambda=10$ | 98.00% | 0.7099 | 100% | 98.54% | 82.46% | 6.38% | 77.19% |
| | $K=5, \lambda=10$ | 98.31% | 0.7076 | 100% | 98.54% | 83.23% | 7.77% | 76.76% |

Therefore, their final performance in stability evaluations is suboptimal. In contrast, the Decompdiff (Guan et al., 2023b) model and Targetdiff (Guan et al., 2023a) model adopt our constraint formula, which fully considers the clash between proteins and molecules during the sampling process. Therefore, the mean clash ratio of these two models is significantly lower than the previously mentioned model. We compare the Decompdiff model with the Targetdiff model using our method, which only significantly reduces the average clash ratio by 4.38% but also minimally impacts the connectivity of the molecules generated by the model, resulting in a sample stability of 80.62%. We have also applied our method to the DiffSBDD model, and the experimental results (see Appendix C.3) indicate that our approach can reduce the clash rate in other models. Additionally, we visualize the two scenarios of clash and non-clash between molecules and proteins (see Appendix C.4).

## 5.3 ABLATION STUDY

Our sampling process algorithm basically consists of three adjustable parameters. In this main text, we mainly analyze the parameter $k$. The impact of restricted sampling time steps and weight parameters $\lambda$ is further clarified in the Appendix C.2. As illustrated in Figure 3, the sequence from the innermost to the outermost dashed line represents the spatial proximity of the single closest protein atom, the three closest protein atoms, and the five closest protein atoms to the ligand atom, respectively. As shown in table 2, we observe that different $k$ values do have different effects on the mean clash ratio. Regardless of the time step, the mean clash ratio decreases most significantly

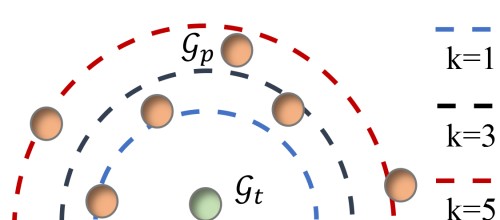

Figure 3: Dotted lines of different colors denote various spatial relationships between molecular atoms and protein atoms.

when k=1. This observation suggests that the protein atoms closest to the ligand atoms play a crucial role in reducing the clash ratio. We find that when $k = 5$ and the constrained time step starts from 300, the connectivity even exceeds the connectivity of the unconstrained sample, reaching 86.00%. This observation suggests that our approach may have the potential to improve connectivity. Taking into account both connectivity and the mean clash ratio, we conclude that the optimal sampling results are achieved when $k = 1$ and the sampling time step begins at 150.

## 5.4 THE PROPERTIES ANALYSIS

Table 4: Comparative analysis of the differences between the sampling outcomes for various constraint time steps and the original sampling results of the model. ↑ indicates that a higher value is preferable, while ↓ signifies that a lower value is more desirable.

| Different Time step | SA ↑ | QED ↑ | LogP | Lipinski ↑ | Vina Score ↓ | High Affinity ↑ |
|---|---|---|---|---|---|---|
| No constraint | 0.597±0.11 | 0.463±0.20 | 1.915±2.84 | 4.450±0.85 | -7.591±2.43 | 53.69% |
| 150-0 | 0.597±0.11 | 0.464±0.20 | 1.884±2.84 | 4.455±0.84 | -7.604±2.43 | 54.23% |
| 300-0 | 0.599±0.11 | 0.464±0.20 | 1.874±2.88 | 4.448±0.85 | -7.633±2.40 | 54.08% |
| 500-0 | 0.598±0.11 | 0.466±0.20 | 1.959±2.85 | 4.459±0.85 | -7.628±2.62 | 54.23% |

**Metrics** We employ widely-used metrics from previous studies (Polykovskiy et al., 2020) to evaluate the impact of our constrained formula on the chemical properties of the generated molecules. (1) **SA** estimates the likelihood that the generated molecules can be synthesized by a chemist (Ertl & Schuffenhauer, 2009). (2) **QED** measures the drug-likeness of the generated molecules (Bickerton et al., 2012). (3) **LogP** not only measures how well a drug will be absorbed, transported, and distributed in the body but also informs how a drug should be formulated and dosed. (4) **Lipinski** indicates whether a generated molecule adheres to Lipinski's 5 rules or not, which is a summary of the experience gained from the existing effective drugs. (5) **Vina score** estimates the binding affinity between the generated molecules and the specific protein. (6) **High affinity** is the percentage of the generated molecules with a Vina score lower than the

Table 3: Jensen-Shannon divergence comparing bond distributions for generated molecules and reference molecules.

| Bond | Raw | 150-0 | 300-0 | 500-0 |
|---|---|---|---|---|
| C-C | 0.3316 | 0.3285 | 0.3271 | 0.3279 |
| C=C | 0.2018 | 0.2089 | 0.2106 | 0.2040 |
| C-N | 0.2494 | 0.2560 | 0.2542 | 0.2505 |
| C=N | 0.1820 | 0.1886 | 0.1865 | 0.2033 |
| C-O | 0.3155 | 0.3152 | 0.3105 | 0.3070 |
| C=O | 0.4101 | 0.4182 | 0.4168 | 0.4204 |
| C:C | 0.2314 | 0.2308 | 0.2283 | 0.2191 |
| C:N | 0.1695 | 0.1314 | 0.1359 | 0.1344 |

original bound ligand of the protein. The SA, QED, LogP, and Lipinski are calculated by the RDkit (Landrum et al., 2022), and the Vina score was calculated by the QVina2 (Alhossary et al., 2015).

**Result Analysis** We added constraint formulas at different sampling steps and generated corresponding 100 molecules for each protein pocket. The experimental results are shown in Table 4. The chemical properties of sampled molecules under different constraint steps are not affected compared to the original sampling results. These experimental results indicate that our constrained formula does not harm the inherent performance of the pre-trained model. Subsequently, we calculated the Jensen-Shannon divergence between the bond length distribution of the generated molecules with different constrained time steps and the bond length distribution of the reference molecules respectively. As shown in Table 3, our method did not destroy the distribution of bonds sampled by the original model. In summary, the whole experimental results demonstrate that our method can solve the clash problem without sacrificing the original performance of the model.

## 6 CONCLUSION

In this study, we introduce a constrained situation aimed at solving the atomic clash issues encountered during the sampling process in SBDD generative diffusion model. We compare the effect with a model that considers clash problems, and the comparison results show that our method can reduce the average clash ratio and improve sampling stability. For future work, we believe that focus on a deeper understanding of the modeling of the diffusion model Spatial relationship between ligands and pockets. This may further improve model stability and efficiency in structure-based drug design.

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

# A    SAMPLING ALGORITHM

---

**Algorithm 1** Sampling Process of PILA

---

**Input:** Sample $\mathcal{G}_t$, $\mathcal{G}_p$, equivariant model $q_\theta$
   Pre-train a structure-based molecular diffusion model by optimizing.
   Initial $\mathcal{G}_T$ from Gaussian Noise
   Set the limit time step $T_l$
   **for** $t = T$ to $0$ **do**
      Predict the $s_\theta(\mathcal{G}_t, \mathcal{G}_p, t)$
      Calculate the $h_l^{t-1} = x_l^t + \epsilon_t s_\theta(\mathcal{G}_t, \mathcal{G}_p, t), \forall t = T, ..., 1.$
      Sample random noise $\epsilon$
      $x_h^{t-1} = h_l^{t-1} + \sqrt{2\epsilon_t} z_{t-1}$
      **if** $0 < t < T_l$ **then**
         $\tilde{x}_h^{t-1} = \text{prox}_g(x_h^{t-1})$
         $x_l^{t-1} = \tilde{x}_h^{t-1}$
      **else**
         $x_l^{t-1} = x_h^{t-1}$
      **end if**
      $\mathcal{G}_{t-1} = [x_l^{t-1}, v_l^{t-1}]$
      Modified the input of $s_\theta$ as $s_\theta(\mathcal{G}_{t-1}, \mathcal{G}_p, t-1)$
   **end for**

---

**Algorithm 2** The Solution Process of Proximal

---

**Input:** $x_h^{t-1}$, $x_p$, $\lambda$, $k$
   **Procedure:**
   Set the require grad attribute of $x_h^{t-1}$ to True.
   Clone a new reference $x_f^{t-1}$ from $x_h^{t-1}$.
   Identify the $k$ protein atoms, denoted as $x_p^k$, that are closest to the ligand atoms.
   Define the optimization function $f = (x_h^{t-1} - x_f^{t-1})^2 + \lambda g(x_h^{t-1})$.
   Define the optimizer as LBFGS.
   Perform an optimizer step.
   **return** $x_h^{t-1}$

---

# B    PROOF OF THEOREM 1

## B.1    ASSUMPTIONS AND DEFINITIONS

**Assumption 1** (LSI). *The target probability distribution $\upsilon$ is supported on $\mathbb{R}^d$ and satisfies LSI with constant $\alpha > 0$, which means for any probability distribution $\rho$ on $\mathbb{R}^d$:*

$$H_\upsilon(\rho) \leq \frac{1}{2\alpha} J_\upsilon(\rho)$$

**Assumption 2** (MGF error assumption). *The error of $s(x)$ has a finite moment generating function of some order $r > 0$ under $\upsilon$:*

$$\epsilon_{mgf}^2 = \epsilon_{mgf}^2(r, s, \upsilon) = \frac{1}{r} \log \mathbb{E}_\upsilon[\exp r \|s(x) - s_\upsilon(x)\|^2] < \infty$$

**Assumption 3** ($L_f$-smoothness). *If $f$ is $L$-smooth for some $0 \leq L_f < \infty$. Thus $\|\nabla f(x) - \nabla f(y)\| \leq L_f \|x - y\|$ for all $x, y \in \mathbb{R}^d$*

**Assumption 4** ($L_s$-Lipschitz). *The score estimator $s$ is $L_s$-Lipschitz for some $0 \leq L_s < \infty$: $\|s(x) - s(y)\| \leq L_s \|x - y\|$*

**Definition 5** (Proximal regularization). *We define $g^{L_g}(x) := prox_g^{\frac{1}{\lambda}}$ as the proximal regularization of non-smooth function $g(x)$. Since $g^{L_g}(x)$ is a proximal regularization, it is gradient Lipschitz with at least $L_g$-Lipschitz, where $L_g := \lambda^{-1}$.*

**Definition 6.** *We define three constants related to $L_f$, $L_s$, and $L_g$ as $L_{s+g}^2 = L_s^2 + L_g^2$, $L_{f+g}^2 = L_f^2 + L_g^2$, $L_{s+g} = L_s + L_g$, and $L_{f+g} = L_f + L_g$.*

## B.2 LEMMAS

**Lemma 2.** *Suppose the assumptions in Theorem 1 hold. Let $\rho_t := \mathrm{Law}(x_t)$ where $x_t$ follows the SDE Eq. 2. Recall Eq. 10 and $g^{L_g}(x)$, we use $\rho$ to replace the $p_\theta(x)$, which means $\rho = \frac{1}{Z_\theta} e^{-U(x)}$. then we have the following bound for the time derivative of KL divergence:*

$$\frac{\partial}{\partial t} H_\upsilon(\rho_t) \leq -\frac{3}{4} J_\upsilon(\rho_t) + \mathbb{E}_{\rho_{0t}}[\|s(x_0) - \nabla g^{L_g}(x_0) - \nabla \log \upsilon(x_t)\|^2],$$

**Lemma 3.** *if the score estimator $s(x_t) \approx -\nabla f(x_t)$ is $L_s$-Lipschitz and $t \leq \frac{1}{3L_s}$, then:*

$$\|(s(x_t) - \nabla g^{L_g}(x_t)) - (s(x_0) - \nabla g^{L_g}(x_0))\|^2$$
$$\leq 18 L_{s+g}^2 t^2 \|s(x_t) + \nabla f(x_t)\|^2 + 18 L_{s+g}^2 t^2 \|\nabla \log \upsilon(x_t)\|^2 + 6 L_{s+g}^2 t \|z_0\|^2$$

**Lemma 4.** *Suppose the assumptions in Theorem 1 hold, then along each step of the sampling algorithm,*

$$H_\upsilon(\rho_{k+1}) \leq e^{-\frac{1}{4}\alpha h} H_\upsilon(\rho_k) + 144 d L_{s+g}^2 L_{f+g} h^3 + 24 d L_{s+g}^2 h^2 + \frac{9}{2} \epsilon_{mgf}^2 h.$$

## B.3 PROOFS

*Proof.* Proof of Lemma 2. The continuity equation corresponding to Eq.2 is

$$\frac{\partial \rho_t(x)}{\partial t} = -\nabla \cdot (\rho_t(x) \mathbb{E}_{\rho_{0|t}}[s(x_0) - \nabla g^{L_g}(x_0)|x_t = x]) + \nabla \rho_t(x)$$

Therefore.

$$\frac{\partial}{\partial t} H_\upsilon(\rho_t) = \int_{\mathbb{R}^d} (-\nabla \cdot (\rho_t(x) \mathbb{E}_{\rho_{0|t}}[s(x_0) - \nabla g^{L_g}(x_0)|x_t = x]) + \nabla \cdot (\rho_t \nabla \log \frac{\rho_t}{\upsilon}) + \nabla \cdot (\rho_t \nabla \log \upsilon)) \log \frac{\rho_t}{\upsilon} dx$$

$$= \int_{\mathbb{R}^d} (\nabla \cdot (\rho_t (\nabla \log \frac{\rho_t}{\upsilon} - \mathbb{E}_{\rho_{0|t}}[s(x_0) - \nabla g^{L_g}(x_0)|x_t = x] + \nabla \log \upsilon))) \log \frac{\rho_t}{\upsilon} dx$$

$$= -\int_{\mathbb{R}^d} \rho_t \langle \log \frac{\rho_t}{\upsilon} - \mathbb{E}_{\rho_{0|t}}[s(x_0) - \nabla g^{L_g}(x_0)|x_t = x] + \nabla \log \upsilon, \log \frac{\rho_t}{\upsilon} \rangle dx$$

$$= -\int_{\mathbb{R}^d} \rho_t \|\nabla \log \frac{\rho_t}{\upsilon}\|^2 dx + \int_{\mathbb{R}^d} \rho_t \langle \mathbb{E}_{\rho_{0|t}}[s(x_0) - \nabla g^{L_g}(x_0)|x_t = x] - \nabla \log \upsilon, \log \frac{\rho_t}{\upsilon} \rangle dx$$

$$= -J_\upsilon(\rho_t) + \int_{\mathbb{R}^d} \rho_t \langle \mathbb{E}_{\rho_{0|t}}[s(x_0) - \nabla g^{L_g}(x_0)|x_t = x] - \nabla \log \upsilon, \log \frac{\rho_t}{\upsilon} \rangle dx$$

$$= -J_\upsilon(\rho_t) + \mathbb{E}_{\rho_{0t}}[\langle s(x_0) - \nabla g^{L_g}(x_0) - \nabla \log \upsilon(x_t), \nabla \log \frac{\rho_t(x_t)}{\upsilon(x_t)} \rangle]$$

$$\leq -J_\upsilon(\rho_t) + \mathbb{E}_{\rho_{0t}}[\|s(x_0) - \nabla g^{L_g}(x_0) - \nabla \log \upsilon(x_t)\|^2] + \frac{1}{4} \mathbb{E}_{\rho_{0t}}[\|\nabla \log \frac{\rho_t(x_t)}{\upsilon(x_t)}\|^2]$$

$$= -J_\upsilon(\rho_t) + \mathbb{E}_{\rho_{0t}}[\|s(x_0) - \nabla g^{L_g}(x_0) - \nabla \log \upsilon(x_t)\|^2] + \frac{1}{4} J_\upsilon(\rho_t)$$

$$= -\frac{3}{4} J_\upsilon(\rho_t) + \mathbb{E}_{\rho_{0t}}[\|s(x_0) - \nabla g^{L_g}(x_0) - \nabla \log \upsilon(x_t)\|^2]$$

$\square$

*Proof.* Proof of Lemma 3. By $L_s$-Lipschitzness of $s$ and $L_g$-smoothness of $g^{L_g}$, where

$$\|s(x_t) - s(x_0)\|^2 \leq L_s^2 \|x_t - x_0\|^2,$$
$$\|\nabla g^{L_g}(x_t) - \nabla g^{L_g}(x_0)\|^2 \leq L_g^2 \|x_t - x_0\|^2$$

and then

$$\|(s(x_t) - \nabla g^{L_g}(x_t)) - (s(x_0) - \nabla g^{L_g}(x_0))\|^2 \le \|s(x_t) - s(x_0)\|^2 + \|\nabla g^{L_g}(x_t) - \nabla g^{L_g}(x_0)\|^2$$
$$\le (L_s^2 + L_g^2)\|x_t - x_0\|^2$$
$$\le L_{s+g}^2 \|ts(x_0) - t\nabla g^{L_g}(x_0) + \sqrt{2t}z_0\|^2$$
$$\le 2L_{s+g}^2 t^2 \|s(x_0) - \nabla g^{L_g}(x_0)\|^2 + 4L_{s+g}^2 t\|z_0\|^2.$$

Let $\mathcal{U}(x) := s(x) - \nabla g^{L_g}(x)$ for the sake of our subsequent analysis, we use a bound in terms of $\mathcal{U}(x_t)$ rather than $\mathcal{U}(x_0)$, Therefore, we opt to utilize

$$L_{s+g}\|x_t - x_0\| \ge \|\mathcal{U}(x_t) - \mathcal{U}(x_0)\| \ge \|\mathcal{U}(x_0)\| - \|\mathcal{U}(x_t)\|$$

Rearranging it gives

$$\|\mathcal{U}(x_0)\| \le L_{s+g}\|x_t - x_0\| + \|\mathcal{U}(x_t)\|$$
$$= L_{s+g}\|ts(x_0) - t\nabla g^{L_g}(x_0) + \sqrt{2t}z_0\| + \|\mathcal{U}(x_t)\|$$
$$\le \frac{1}{3}\|\mathcal{U}(x_0)\| + L_{s+g}\sqrt{2t}\|z_0\| + \|\mathcal{U}(x_t)\| \quad \text{since } t \le \frac{1}{3L_{s+g}}$$

It follows that

$$\|\mathcal{U}(x_0)\| \le \frac{3}{2}\|\mathcal{U}(x_t)\| + \frac{3}{\sqrt{2}}L_{s+g}\sqrt{t}\|z_0\| \to \|\mathcal{U}(x_0)\|^2 \le \frac{9}{2}\|\mathcal{U}(x_t)\|^2 + 9L_{s+g}^2 t\|z_0\|^2$$

So we can bound $\|\mathcal{U}(x_t) - \mathcal{U}(x_0)\|^2$ as follows:

$$\|\mathcal{U}(x_t) - \mathcal{U}(x_0)\|^2 \le 2L_{s+g}^2 t^2 \|\mathcal{U}(x_0)\|^2 + 4L_{s+g}^2 t\|z_0\|^2$$
$$\le 2L_{s+g}^2 t^2 (\frac{9}{2}\|\mathcal{U}(x_t)\|^2 + 9L_{s+g}^2 t\|z_0\|^2) + 4L_{s+g}^2 t\|z_0\|^2$$
$$= 9L_{s+g}^2 t^2 \|\mathcal{U}(x_t)\|^2 + (18L_{s+g}^4 t^3 + 4L_{s+g}^2 t)\|z_0\|^2$$
$$\le 9L_{s+g}^2 t^2 \|\mathcal{U}(x_t)\|^2 + 6L_{s+g}^2 t\|z_0\|^2$$
$$= 9L_{s+g}^2 t^2 \|s(x_t) - \nabla g^{L_g}(x_t) - \nabla f(x_t) + \nabla f(x_t))\| + 6L_{s+g}^2 t\|z_0\|^2$$
$$\le 18L_{s+g}^2 t^2 \|s(x_t) + \nabla f(x_t)\| + 18L_{s+g}^2 t^2 \|\nabla \log \upsilon(x_t)\| + 6L_{s+g}^2 t\|z_0\|^2,$$

where $\nabla \log \upsilon(x_t) = -(\nabla f(x_t) + \nabla g^{L_g}(x_t))$. $\qquad\square$

*Proof.* (Proof of Lemma 4). Let $M(x) = \|s(x_t) + \nabla f(x_t)\|^2$, By Lemma 2,

$$\frac{\partial}{\partial t} H_\upsilon(\rho_t) \le -\frac{3}{4}J_\upsilon(\rho_t) + \mathbb{E}_{\rho_{0t}}[\|\mathcal{U}(x_0) - \nabla \log \upsilon(x_t)\|^2]$$
$$\le -\frac{3}{4}J_\upsilon(\rho_t) + 2\mathbb{E}_{\rho_{0t}}[\|\mathcal{U}(x_0) - \mathcal{U}(x_t)\|^2] + 2\mathbb{E}_{\rho_{0t}}[\|\mathcal{U}(x_t) - \nabla \log \upsilon(x_t)\|^2]$$
$$\le -\frac{3}{4}J_\upsilon(\rho_t) + 2\mathbb{E}_{\rho_{0t}}[18M(x)\| + 18L_{s+g}^2 t^2 \|\nabla \log \upsilon(x_t)\| + 6L_s^2 t\|z_0\|^2] + 2\mathbb{E}_{\rho_t}[M(x)]$$
$$= -\frac{3}{4}J_\upsilon(\rho_t) + (36L_{s+g}^2 t^2 + 2)\mathbb{E}_{\rho_t}[M(x)] + 36L_{s+g}^2 t^2 \mathbb{E}_{\rho_t}[\|\nabla \log \upsilon(x_t)\|^2] + 12dL_{s+g}^2 t$$
$$\le -\frac{3}{4}J_\upsilon(\rho_t) + \frac{9}{4}\mathbb{E}_{\rho_t}[M(x)] + 36L_s^2 t^2 \mathbb{E}_{\rho_t}[\|\nabla \log \upsilon(x_t)\|^2] + 12dL_{s+g}^2 t$$
$$\quad \text{since } t^2 \le h^2 \le \frac{\alpha^2}{144L_{s+g}^2 L_{f+g}^2} \le \frac{1}{144L_{f+g}}$$
$$\le -\frac{3}{4}J_\upsilon(\rho_t) + \frac{9}{4}\mathbb{E}_{\rho_t}[M(x)] + 36L_{s+g}^2 t^2 (\frac{4L^2}{\alpha}H_\upsilon(\rho_t) + 2dL_{f+g}) + 12dL_{s+g}^2 t$$
$$= -\frac{3}{4}J_\upsilon(\rho_t) + \frac{9}{4}\mathbb{E}_{\rho_t}[M(x)] + \frac{144L_{s+g}^2 t^2 L^2}{H_\upsilon(\rho_t)} + 72dL_{s+g}^2 t^2 L_{f+g} + 12dL_{s+g}^2 t$$
$$\le -\frac{3}{4}J_\upsilon(\rho_t) + \frac{9}{4}\mathbb{E}_{\rho_t}[M(x)] + \alpha H_\upsilon(\rho_t) + 72dL_{s+g}^2 t^2 L_{f+g} + 12dL_{s+g}^2 t \quad \text{since } t^2 \le h^2 \le \frac{\alpha^2}{144L_{s+g}^2 L_{f+g}^2}$$
$$\le -\frac{1}{2}\alpha H_\upsilon(\rho_t) + \frac{9}{4}\mathbb{E}_{\rho_t}[M(x)] + 72dL_{s+g}^2 t^2 L_{f+g} + 12dL_{s+g}^2 t \quad \text{by } \alpha\text{-LSI}$$

where the second term can be bounded as follows:

$$\mathbb{E}_{\rho_t}[M(x)] \le \epsilon_{mgf}^2 + \frac{\alpha}{9} H_\upsilon(\rho_t)$$

so we have:

$$
\begin{aligned}
\frac{\partial}{\partial t} H_\upsilon(\rho_t) &\le -\frac{1}{4}\alpha H_\upsilon(\rho_t) + 72dL_{s+g}^2 t^2 L_{f+g} + 12dL_{s+g}^2 t + \frac{9}{4}\epsilon_{mgf}^2 \\
&\le -\frac{1}{4}\alpha H_\upsilon(\rho_t) + 72dL_{s+g}^2 h^2 L_{f+g} + 12dL_{s+g}^2 h + \frac{9}{4}\epsilon_{mgf}^2 \quad \text{since } t \in (0, h).
\end{aligned}
$$

This is equivalent to:

$$\frac{\partial}{\partial t} e^{-\frac{1}{4}\alpha t} H_\upsilon(\rho_t) \le e^{-\frac{1}{4}\alpha t}(72dL_{s+g}^2 h^2 L_{f+g} + 12dL_{s+g}^2 h + \frac{9}{4}\epsilon_{mgf}^2)$$

Therefore,

$$
\begin{aligned}
H_\upsilon(\rho_h) &\le e^{-\frac{1}{4}\alpha h} H_\upsilon(\rho_0) + e^{\frac{1}{4}\alpha h}\frac{4e^{\frac{1}{4}\alpha h} - 1}{\alpha}(72dL_{s+g}^2 h^2 L_{f+g} + 12dL_{s+g}^2 h + \frac{9}{4}\epsilon_{mgf}^2) \\
&\le e^{-\frac{1}{4}\alpha h} H_\upsilon(\rho_0) + 2h(72dL_{s+g}^2 h^2 L_{f+g} + 12dL_{s+g}^2 h + \frac{9}{4}\epsilon_{mgf}^2)
\end{aligned}
$$

where the $e^{\frac{1}{4}\alpha h} \le 1$ and $e^c - 1 \le 2c$ for $c = \frac{1}{4}\alpha h \in (0, 1)$. Substitute the $\rho_0$ as $\rho_k$ and $\rho_h$ as $\rho_{k+1}$, we can get the desired bound:

$$H_\upsilon(\rho_{k+1}) \le e^{-\frac{1}{4}\alpha h} H_\upsilon(\rho_k) + 144dL_{s+g}^2 h^3 L_{f+g} + 24dL_{s+g}^2 h^2 + \frac{9}{2}\epsilon_{mgf}^2$$

$\square$

*Proof of Theorem 1.* Applying the recursion contraction in Lemma 4 $k$ times, we have

$$
\begin{aligned}
H_\upsilon(\rho_{k+1}) &\le e^{-\frac{1}{4}\alpha h k} H_\upsilon(\rho_0) + \sum_{i=0}^{k} e^{-\frac{1}{4}\alpha h i}\left(144dL_{s+g}^2 h^3 L_{f+g} + 24dL_{s+g}^2 h^2 + \frac{9}{2}\epsilon_{mgf}^2\right) \\
&\overset{(i)}{\le} e^{\frac{1}{4}\alpha h k} H_\upsilon(\rho_0) + \frac{1}{1 - e^{-\frac{1}{4}\alpha h}}\left(144dL_{s+g}^2 h^3 L_{f+g} + 24dL_{s+g}^2 h^2 + \frac{9}{2}\epsilon_{mgf}^2\right) \\
&\le e^{\frac{1}{4}\alpha h k} H_\upsilon(\rho_0) + \frac{16}{3\alpha h}\left(144dL_{s+g}^2 h^3 L_{f+g} + 24dL_{s+g}^2 h^2 + \frac{9}{2}\epsilon_{mgf}^2\right) \\
&\le e^{\frac{1}{4}\alpha h k} H_\upsilon(\rho_0) + \frac{768d(L_{s+g})^2 L_{f+g}}{\alpha}h^2 + \frac{128d(L_{s+g})^2}{\alpha}h + \frac{8}{3\alpha}\epsilon_{mgf}^2 \quad \text{since } L_{s+g}^2 \le (L_{s+g})^2 \\
&\le e^{\frac{1}{4}\alpha h k} H_\upsilon(\rho_0) + \frac{128d(L_{s+g})}{\alpha}(L_{s+g} + L_{f+g})dh + \frac{8}{3\alpha}\epsilon_{mgf}^2 \quad \text{since } h < \frac{\alpha}{12L_{s+g}L_{f+g}} \le \frac{1}{12L_{s+g}},
\end{aligned}
$$

where $(i)$ employs the inequality $1 - e^{-c} \ge \frac{3}{4}c$ for $0 < c = \frac{1}{4}\alpha h \le \frac{1}{4}$, which holdes since $h \le \frac{1}{2}\alpha$. $\square$

### B.4 PROBLEM OF DECOMPDIFF

As we mentioned in the paper, DecompDiff (Guan et al., 2023b) employ an maximum in the gradient of constraints which make the function to be non-smooth. Moreover, when we resolve $S(x)$ back into $p_\theta(x)$, the tractable distribution $p_\theta(x)$ becomes:

$$p'_\theta(x) = e^{\max(0, \gamma - S(x))} p_\theta(x),$$

which requires both $\int e^{\max(0, \gamma - S(x))} = 1$ and $\int p_\theta(x) = 1$. However, the truncated function $\int e^{\max(0, \gamma - S(x))}$ does not satisfy the requirement although $\int e^{-S(x)} = 1$ stands. we have incorporated the gradient guidance method (Decompdiff) into the targetdiff model and evaluated its performance. As depicted in Table 5, this method also succeeds in reducing the clash ratio. Moreover, our method demonstrates superior stability compared to Decompdiff, given the nearly equal mean clash ratios.

Table 5: The comparsion between Gradient guidance (Decompdiff) and Proximal (ours).

| Method | Validity ↑ | Diversity ↑ | Novelty ↑ | Uniqueness ↑ | Connectivity ↑ | Mean CR ↓ | Stability ↑ |
|---|---|---|---|---|---|---|---|
| Raw | 98.31% | 0.7107 | 100% | 98.77% | 85.38% | 12.31% | 74.86% |
| Gradient | 94.31% | 0.6504 | 100% | 99.39% | 74.46% | 1.86% | 73.08% |
| Proximal | 98.15% | 0.7093 | 100% | 99.77% | 76.62% | 1.92% | **75.15%** |

## C  EXPERIMENTS

### C.1  2 Å MOTIVATION

The threshold we set at 2Å is not arbitrary. Firstly, from a chemical standpoint, when two atoms are sufficiently close, a bond forms between them. Secondly, the statistical pair distribution analysis reveals that there are virtually no pair distances smaller than 2Å in the crossdocked training dataset. Thirdly, the statistical results of the real dataset PDBbind (as shown in Figure 4) indicate that most pairing distances are indeed less than 2Å. Lastly, PoseCheck also tackles the steric clash issue, defining pairwise distances between molecular atoms and protein atoms in a manner similar to our definition. Their definition slightly deviates from ours, as they set a threshold of 0.5Å based on bond lengths. In summary, considering the factors mentioned above, we establish the threshold at 2Å..

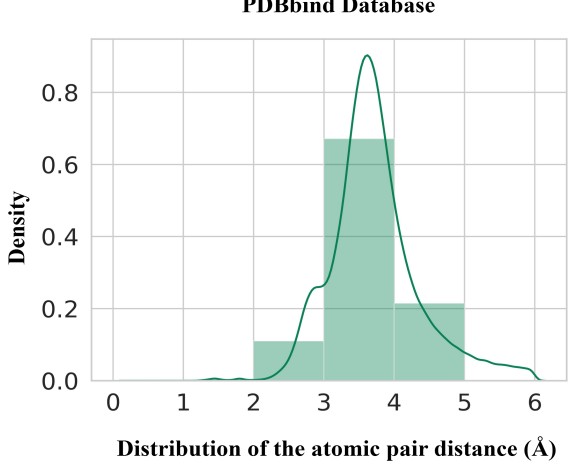

Figure 4: Statistical distribution of pairwise distances between proteins and molecules within 6Å in the PDBbind Dataset.

### C.2  ANALYSIS OF CONSTRAINED TIME STEPS AND WEIGHT PARAMETERS

To identify the optimal combination of sampling time steps and weight, we conducted multiple ablation experiments. As illustrated in Table 6, a very low clash rate can be obtained when the weight is generally 10, given the same time step and k value. Simultaneously, under the same time step and coefficient, a larger $k$ value results in a more significant impact on the clash rate. It can be observed that the best clash results are achieved when $k = 5$ and $\lambda = 100$. Under the same $k$ value and weight, the later the time constraint step, the higher the stability value. Consequently, considering the overall sampling stability, the optimal results were obtained when k=1, $\lambda = 10$, and the sampling constraint time step ranged from 150 to 0. We assess the average time (as shown in Table 7) needed to sample an individual molecule at different constrained time steps. The time consumption related to this procedure indicates that our approach is suitable for large-scale sampling.

Table 6: The impact of different parameters on the final sampling results.

| Time | Influence Factor | Validity ↑ | Diversity ↑ | Novelty ↑ | Uniqueness ↑ | Connectivity ↑ | Mean CR ↓ | Stability ↑ |
|------|------------------|-----------|-------------|-----------|--------------|----------------|-----------|-------------|
|       | $K=1, \lambda=1$   | 98.15% | 0.7114 | 100% | 98.69% | 85.54% | 8.69% | 78.10% |
|       | $K=1, \lambda=10$  | 98.15% | 0.7117 | 100% | 98.69% | 84.31% | 4.38% | 80.62% |
| 150-0 | $K=1, \lambda=100$ | 97.85% | 0.7124 | 100% | 98.85% | 80.31% | 5.38% | 75.99% |
|       | $K=3, \lambda=100$ | 98.00% | 0.7115 | 100% | 98.69% | 80.00% | 4.46% | 76.43% |
|       | $K=5, \lambda=100$ | 97.69% | 0.7116 | 100% | 98.69% | 80.31% | 3.15% | 77.78% |
|       | $K=1, \lambda=10$  | 97.69% | 0.7109 | 100% | 98.69% | 82.15% | 4.31% | 78.61% |
| 300-0 | $K=1, \lambda=100$ | 98.31% | 0.7082 | 100% | 99.08% | 81.08% | 3.54% | 78.20% |
|       | $K=3, \lambda=100$ | 98.15% | 0.7112 | 100% | 99.15% | 80.62% | 3.85% | 77.51% |
|       | $K=5, \lambda=100$ | 98.00% | 0.7106 | 100% | 99.08% | 79.85% | 2.23% | 78.07% |
|       | $K=1, \lambda=10$  | 98.15% | 0.7062 | 100% | 98.77% | 79.38% | 3.23% | 76.82% |
| 500-0 | $K=1, \lambda=100$ | 97.54% | 0.7105 | 100% | 99.08% | 71.69% | 4.46% | 68.49% |
|       | $K=3, \lambda=100$ | 98.31% | 0.7081 | 100% | 98.92% | 74.77% | 2.00% | 73.27% |
|       | $K=5, \lambda=100$ | 98.15% | 0.7093 | 100% | 99.77% | 76.62% | 1.92% | 75.15% |

Table 7: The comparison of time consumption across various constrained time steps.

|          | Raw sample | 500-0 | 300-0 | 150-0 |
|----------|-----------|-------|-------|-------|
| Time($s$) | 17.30 | 22.28 | 18.03 | 18.40 |

## C.3 THE EXPERIMENTAL RESULTS OF THE DIFFSBDD

We apply our method to the DiffSBDD model, and the outcomes (as shown in Table 8) indicate the effectiveness of our approach in reducing the clash ratio across various models.

Table 8: The experimental results of the DiffSBDD mdoel.

| Time | Influence Factor | Validity ↑ | Diversity ↑ | Novelty ↑ | Uniqueness ↑ | Mean CR ↓ |
|------|------------------|-----------|-------------|-----------|--------------|-----------|
|       | No constrained     | 100%   | 0.8246 | 100% | 97.44% | 17.17% |
| 500-0 | $K=1, \lambda=10$ | 99.78% | 0.8277 | 100% | 96.88% | 9.64%  |
| 300-0 | $K=1, \lambda=10$ | 100%   | 0.8284 | 100% | 96.88% | 9.95%  |
| 150-0 | $K=1, \lambda=10$ | 100%   | 0.8263 | 100% | 96.76% | 10.37% |

## C.4 THE VISUALIZATION RESULTS OF ADDING CONSTRAINED FORMULA

we select three proteins that have the clash situation, and then we sample the molecules for each of the proteins. we design a dual-sample process, the model can generate the clash molecules and no clash molecules at the same time. As shown in Figure 5, before constrained we can observe that those atoms pair between molecular atoms and residue atoms are too close, and after adding the constrained the atoms of molecules are pushed away from the residues atoms.

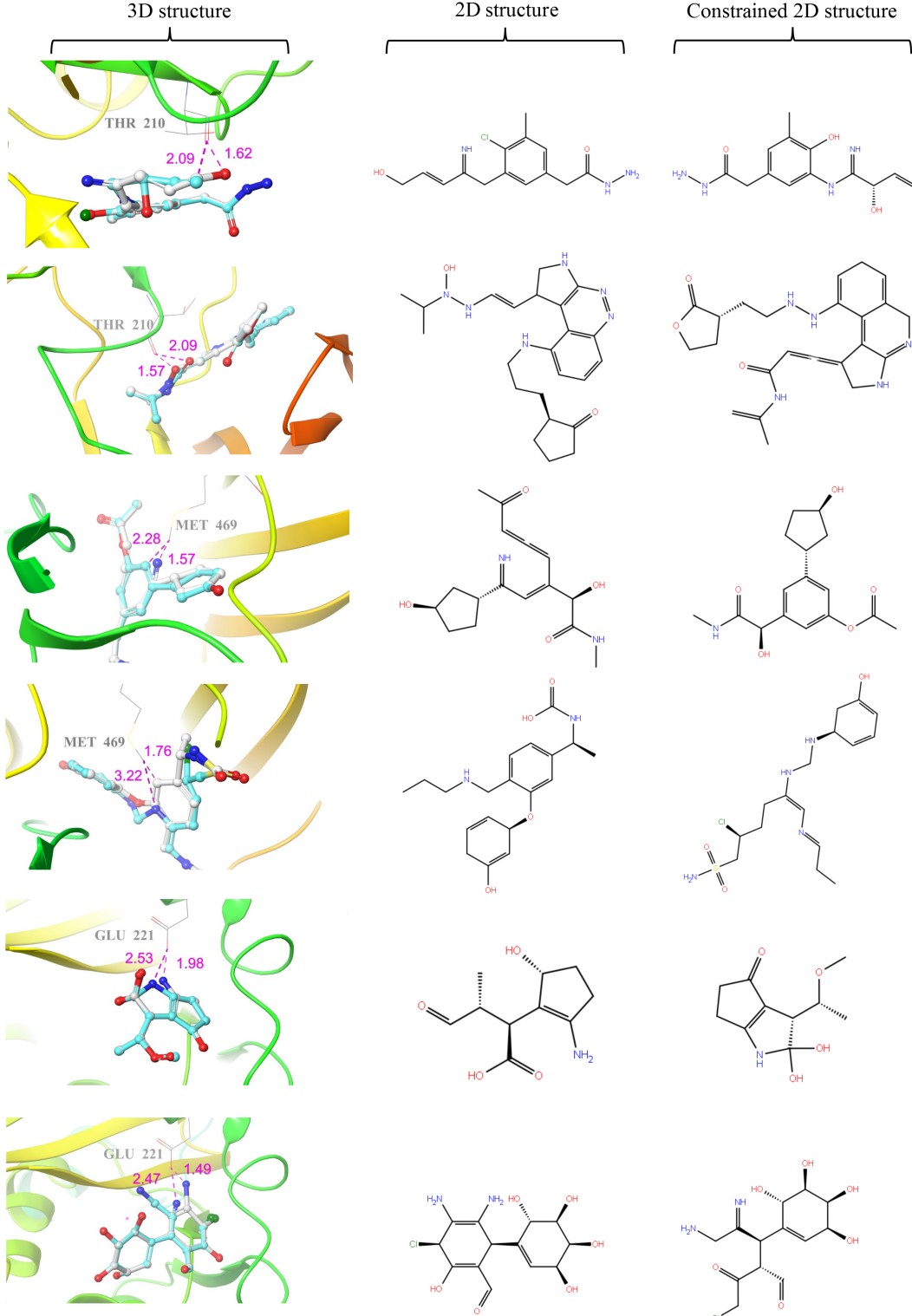

Figure 5: **3D structure** shows the coexistence of constrained and unconstrained molecules in a protein pocket. **2D structure** and **Constrained 2D structure** display the 2D structures of generated molecules and generated constrained molecules respectively.

