# OpenReview forum: "Reducing Atomic Clashes in Geometric Diffusion Models for 3D Structure-Based Drug Design"
_ICLR.cc/2024/Conference — Submitted to ICLR 2024_

### Official Review · Reviewer_2tn2 · 2023-10-24

**Soundness:** 2 fair
**Presentation:** 2 fair
**Contribution:** 2 fair
**Rating:** 3
**Confidence:** 3

**Summary:**

The authors noticed the problem of spatial clash between the generated molecule and protein in existing generative methods. They proposed to formulate it as a constrained optimization problem and solve it by modifying the sampling process employing a proximal operator based on a diffusion model. They also provided a theoretical justification of their method. They compared their method with state-of-the-art generative methods, and their experiments show that the clash ratio is effectively reduced by employing this technique.

**Strengths:**

The clash ratio is effective reduced with the proposed technique.

**Weaknesses:**

1. Novelty: The authors are not the first to discuss the problem of spatial clash. The novelty is limited.
2. Biological Motivation: What is the motivation behind constraining the minimal distance between the protein and the generated molecules to be more than 2Å - a constant? Is it assuming that this distance is not conditioned on the properties of the protein and the generated molecules?
3. Biological Motivation Again: Why 2Å is an appropriate threshold? Is there any underlying biological insight? It is true that this threshold fits well on the Crossdocked2020 dataset, does it still hold for other datasets?
4. Performance: The performance gain on most benchmarks seems to be marginal, e.g. validity, novelty, uniqueness, diversity, SA, QED, Lipinski, and bond distributions.
5. Clarity: The authors mentioned Van der Waals force and Hydrogen Bonding in their abstract. Can their model better capture these interactions?

**Questions:**

N/A

---

> ### Author Response · Authors · 2023-11-22
> **Rebuttal**
>
> Thank you for taking the time to read and review our paper! Please see our responses below. We have started working on the corresponding changes. I hope that our response has sufficiently addressed your inquiry. All our modifications have been distinctly marked in blue font within the main body of the text.
>
> **W1**: Novelty: Both PoseCheck and Decomp have recognized clash issues. However, the former does not offer a solution, while the latter's approach has certain drawbacks, which we have highlighted in the theoretical issues section of our paper's appendix B.4. We also implemented their methods in TargetDiff for comparison. The experimental outcomes demonstrate that our method attains greater stability while reducing clash to a similar degree as theirs, suggesting that our approach is more effective in addressing clash problems.
>
> **W2, W3**: 2Å Motivation: The threshold we set at 2Å is not arbitrary. Firstly, from a chemical standpoint, when two atoms are sufficiently close, a bond forms between them. Secondly, the statistical pair distribution analysis reveals that there are virtually no pair distances smaller than 2Å in the crossdocked training dataset. Thirdly, the statistical results of the real dataset PDBbind (as shown in Appendix C.1) indicate that most pairing distances are indeed more than 2Å. Lastly, PoseCheck also tackles the steric clash issue, defining pairwise distances between molecular atoms and protein atoms in a manner similar to our definition. Their definition slightly deviates from ours, as they set a threshold of 0.5Å based on bond lengths. In summary, considering the factors mentioned above, we establish the threshold at 2Å.
>
> **W4**: Performance: Our main objective is to minimize clash while maintaining the performance of other metrics unaffected by this reduction. We have carried out a series of benchmark experiments, encompassing validity, novelty, uniqueness, diversity, SA, QED, and Lipinski. The experimental outcomes suggest that our approach does not diminish the model's original generative abilities and efficiently reduces unreasonable sampling.
>
> **W5**: Clarity: We have highlighted the significance of Van der Waals forces and Hydrogen Bonding because we believe that the pairwise distance between atoms is crucial for these two types of forces. These forces can effectively manifest only when the atoms are at an appropriate distance from each other.

---

> > ### Comment · Reviewer_2tn2 · 2023-11-23
> > **Thank the authors for the rebuttal**
> >
> > W2, W3: I see the intuition of the authors' work, and I see the soundness in a statistical sense, but I still don't think the method is sound in a biological sense.
> >
> > W5: Can authors justify this claim? (e.g. in experiments)
> >
> > For the reasons above, I keep my score.

---

> > > ### Author Response · Authors · 2023-11-23
> > > **Rebuttal2**
> > >
> > > We thank your feedback!
> > >
> > > **W2, W3**: First of all, considering the biological aspect, Hooft et al. pointed out that the existence of clash in biological information data itself cannot be ignored. Therefore, it is meaningful and improtant to avoid the occurrence of clash during the molecule generation process. Secondly, in terms of machine learning, we integrated the proximal Langevin algorithm (PLA) method into the molecular diffusion model sampling process with a robust theoretical analysis of convergence, which make a substantial contribution to the field of machine learning. ref: https://doi.org/10.1038/381272a0
> > >
> > > **W5**: The distance is indeed a critical factor for these two types of forces. Firstly, in all tested models for molecule generation, hydrogen bonds are not considered during the training phase. This is because their positions can be easily deduced once the skeleton of the molecule is known. Second, the formula representing van der Waals forces is $F = -\frac{{2\mu _1^2\mu _2^2}}{{3kT{r^6}}} \cdot \frac{1}{{{{(4\pi{\epsilon_0})}^2}}} - \frac{{\mu _1^2{\alpha _2}}}{{{{(4\pi{\varepsilon _0})}^2}{r^6}}} - \frac{3}{2}\frac{{{I_1}{I_2}}}{{{I_1} + {I_2}}}(\frac{{{\alpha _1}{\alpha _2}}}{{{r^6}}})\frac{1}{{{{(4\pi{\epsilon_0})}^2}}}$, where $r$ represents the distance between the centers of mass of the two molecules, $\mu_1$ and $\mu_2$ are the dipole moments of the molecules, $\epsilon_0$ is the permittivity of free space, $k$ is the Boltzmann constant, and $T$ stands for the thermodynamic temperature. $I_1$ and $I_2$ indicate the ionization energies of the molecules, while $\alpha_1$ and $\alpha_2$ refer to their polarizabilities. It is clear from this formula that the strength of these forces really depends onthe distance between molecules. Consequently, by applying our constraints to reduce distance clash between atoms, we ensure that the generated molecules maintain  reasonable van der Waals forces within the atoms between the molecules and the proteins.

---

### Official Review · Reviewer_7c1E · 2023-10-29

**Soundness:** 2 fair
**Presentation:** 2 fair
**Contribution:** 1 poor
**Rating:** 3
**Confidence:** 4

**Summary:**

There are some intermolecular atomic clashes between the generated ligand molecule and protein pockets. To tackle this issue, the authors of the paper propose a new sampling process specifically designed to prevent such unwanted collisions. This is accomplished by integrating a non-convex constraint within the current Langevin Dynamics (LD) of GDM, aided by the use of proximal regularization techniques. This new process forces molecular coordinates to adhere to set physical constraints. Crucially, the proposed method doesn't require any changes to be made to the training process of GDMs. Empirical evaluations have shown that this method significantly reduces atomic clashes compared to the original LD process of GDMs.

**Strengths:**

1. The problem formulation and presentation are clear.
2. The method is rationale and theoretical analysis is provided.
3. The proposed method improves the connectivity, mean CR, and stability of generated ligand molecules.

**Weaknesses:**

1. This method is not novel to some extent, which is some kind of similar to the validity guidance proposed in DecompDiff.
2. The experimental results on molecular properties of generated ligands are not satisfying. The improvement on Vina score is only marginal. It is notable that vina score is one of the most metrics for structure-based drug design and strongly related to intermolecular spatial interaction between pockets and generated ligands.

**Questions:**

Can the author provide more comparison between the validity guidance proposed in DecompDiff and proximal sampling proposed in this paper? Can the vina score be further improved?

---

> ### Author Response · Authors · 2023-11-22
> **Rebuttal**
>
> Thank you for taking the time to read and review our paper! Please see our responses below. We have started working on the corresponding changes. I hope that our response has sufficiently addressed your inquiry. All our modifications have been distinctly marked in blue font within the main body of the text.
>
> **W1, Q1**: Compare:  In response to the reviewer's suggestion, we have integrated the gradient guidance method (Decompdiff) into the targetdiff model and assessed its performance. As illustrated in Table 1, this approach also effectively reduces the clash ratio. Furthermore, our method exhibits greater stability compared to Decompdiff, as indicated by the nearly equal average clash ratios. We provide additional details on the theoretical limitations of Decompdiff in Appendix B.4. Our strategy employs the proximal Langevin algorithm (PLA) for conducting inference on diffusion models. Since the sampling procedure for these models involves the application of an inexact Langevin algorithm (ILA), we first present a convergence analysis for the proximal inexact Langevin algorithm (PILA) within the context of sampling. In contrast, the Decompdiff method lacks such theoretical guarantee.
> | Method | Validity | Diversity | Novelty | Uniqueness | Connectivity | Mean CR | Stability |
> |:----------:|:---------:|:------------:|:----------:|:----------------:|:----------------:|:-------------:|:----------:|
> | Raw      |  98.31%	| 0.7107 |	100%   |	98.77%       |	85.38%     |    12.31% |	74.86%|
> | Gradient(Decomp) |94.31%	| 0.6504 |	100% | 99.39% | 74.46%	 | 1.86%  | 73.08% |
> |Proximal(ours)	|98.15%	| 0.7093 |	100% |99.77%	 | 76.62%	 | 1.92%  | 75.15% |
>
> **W2**: Vina score: We posit that the Vina score has a strong correlation with the data distribution learned by the model. On one hand, our primary objective is to minimize clash, and our experimental outcomes convincingly show that our approach accomplishes this by decreasing the average clash ratio while maintaining the distribution of most generated molecules. On the other hand, we consider the observed marginal improvement in the Vina score to be in line with expectations, as it aligns with the degree of clash ratio reduction.

---

### Official Review · Reviewer_F9wF · 2023-10-29

**Soundness:** 2 fair
**Presentation:** 2 fair
**Contribution:** 2 fair
**Rating:** 6
**Confidence:** 4

**Summary:**

The authors introduce a technique that is applied during inference to minimize the of atomic steric clashes when generating molecules within a 3D protein pocket. The proposed method doesn't require retraining the diffusion model as it only involves changing the sampling process during inference. The authors benchmark their method against existing techniques like DiffSBDD and PMDM, demonstrating that their approach yields fewer atomic clashes without compromising other properties such as binding affinity, synthetic accessibility etc. Overall, this work effectively tackles the issue of steric atomic clashes in generative diffusion models within the context of SBDD.

**Strengths:**

1. The authors clearly identifies steric atomic clashes (ligand atoms too close to protein atoms) as a major shortcoming of current geometric diffusion models for structure based drug design. The authors provide strong evidence that this violates physical principles such as Van der Waals force and Hydrogen Bonding.

2. The authors introduces a new proximal constrained sampling technique to reduce atomic clashes, the usage of proximal regularization handles the non-convex distance constraints elegantly. This approach is novel in the context of structure based diffusion models for drug design.

3. The proposed method only modifies the sampling process (at inference time), requiring no changes to model training. This makes the approach easy to implement on top of existing pre-trained diffusion models. The experimental results demonstrate effectiveness using TargetDiff without modification to its training.

**Weaknesses:**

1. The method is only evaluated on one dataset (CrossDock2020) with a single model (TargetDiff). Unclear if the benefits generalize to other datasets (PDBBind) and diffusion models (DiffSBDD).

2. Proximal constrained sampling adds extra computational costs during inference. Could be prohibitively expensive for large datasets or real-time usage and there is no extensive discussion of added computational cost in the paper.

**Questions:**

1. The authors note the proximal constrained sampling adds computational overhead. Were there any optimizations considered to improve efficiency and make the approach scale well to larger datasets? What is the impact on runtime?

2.The authors briefly discuss binding affinity (Vina scores) in their results. However,  this results section is not extensive. A more thorough evaluation could include a case study where molecules generated by the proposed method are compared against an experimental xtal structure. Assessing the 3D overlap between the generated molecules and the ligand bound xtal structure would provide valuable insights.

3. Could the proposed sampling technique be extended to other diffusion-based docking algorithms, such as DiffDock?

### Typos:

There are several typos in the paper. For example, "tTargetDiff", "2Abetween" in the Introduction.

---

> ### Author Response · Authors · 2023-11-22
> **Rebuttal**
>
> Thank you for taking the time to read and review our paper! Please see our responses below. We have started working on the corresponding changes. I hope that our response has sufficiently addressed your inquiry. All our modifications have been distinctly marked in blue font within the main body of the text.
>
> **W1**: Other dataset: We use TargetDiff as our base model without retraining it. Since TargetDiff is trained exclusively on CrossDock, we only tested on CrossDock to ensure a fair comparison. On the other hand, we tested our framework on DiffSBDD, another diffusion-based molecular generative model. See the table below for the results of DiffSBDD.
> | Time | Influence | Validity | Diversity | Novelty | Uniqueness | Mean CR |
> |:------:|:------------:|:----------:|:-----------:|:----------:|:----------------:|:-------------:|
> |         | normal     |	100% |  0.8246   | 100%	  |  97.44%	| 17.17%    |
> | 300   | K=1, b=10 |	99.78%  |	0.8277 | 100%	 | 96.82%	| 9.64% |
> | 200   | K=1, b=10 |100%     |	0.8284 |100%	 | 96.88%	 | 9.95% |
> | 150   | K=1, b=10 |100%     |	0.8263 |100%	 |  96.76% |10.37% |
>
> **W2, Q1**: Computational cost: In the process of proximal sampling, the optimization of the solution is carried out using the LBFGS solver. This solver exhibits a second-order convergence rate, signifying that the time required for the solution process is minimal (as shown in the table below). Moreover, as the atomic position sampled at time t is already in proximity to the optimal solution we aim to obtain, both the number of iterations and the time spent on them are considerably reduced.
> | Targetdiff | Raw | 500-0 | 300-0 | 150-0 |
> |:------------:|:------:|:--------:|:-------:|:--------:|
> |Time(s)   |17.30 | 22.28 | 18.03 | 18.40 |
>
> **Q2**: Larger dataset: We are not entirely certain about the reviewer's inquiry concerning the 'larger dataset'. In real-world applications, sampling is usually performed on a single protein. As displayed in the table associated with question W2 above， we have evaluated the average time needed to sample an individual molecule. The time consumption associated with this process indicates that our approch can be employed for large-scale sampling.
>
> **Q3**: Case study: In Appendix C.3, we have provided a visual analysis of the results between the clash and non-clash scenarios, demonstrating the effectiveness of our approach.
>
> **Q4**: Diffdock：In the docking procedure using DIFFDOCK, the inputs consist of the molecule's random conformations and the protein structure. The molecule undergoes overall rotations and translations, followed by dihedral angle adjustments to achieve a proper fit between the molecular structure and the protein pocket. Conversely, our approach on clash caculation directly works with the molecule's 3D coordinates. As a result, applying our method to DIFFDOCK is not trivial.

---

> > ### Comment · Reviewer_F9wF · 2023-11-22
> >
> > Thanks for addressing the questions and running experiments with DiffSBDD. I really appreciate it. I'm raising my score to 6.

---

### Official Review · Reviewer_xcbL · 2023-10-31

**Soundness:** 2 fair
**Presentation:** 1 poor
**Contribution:** 2 fair
**Rating:** 3
**Confidence:** 3

**Summary:**

In this paper, the authors propose to augment one pocket-conditioned ligand generation (ie TargetDiff) by introducing extra constraints to the Langevin MCMC steps. In particular, the proposed constraint aims at reducing the number of steric clashes of generated molecules. The proposed constraint is not used during training and can be applied on sampling on the fly. The approach is compared to some baselines on the task CrossDocket2020 dataset.

**Strengths:**

- The main idea of the paper—reduce the number of clashes on molecules generated by diffusion-based models—is intuitive and well-grounded. Diffusion-based generative models are known to oversee clashes when generating models.
- The proposed approach is only applied during sampling stage, making it applicable to different models based on the score function.
- The results on the reported metrics are good, but I am not convinced those are the right metrics (see below).

**Weaknesses:**

- The paper is not well written and hard to follow. There are many terms that are mentioned without any definition and some notations are wrong.
- The paper also lacks many details of the implementation and make reproduction difficult.
- The authors uses a cut-off of 2A to consider clash vs non-clash. This seems a very ad-hoc choice. Why did the authors use this number? PoseCheck (Harris et al23) consider a threshold of .5A, which sounds more reasonable.
- There is not much detail in how clash is computed. Since the main contribution of the paper is to reduce the number of clashes, it could be nice to have more details about this metric, how the numbers would change if you change the definition of clash used, etc.
- The choice of evaluation metrics made by the authors are not the best. I am really confused why on Table 1, they show Hoogeboom et al 22 metrics (a model that deals with unconditional generation). These metrics are far from the best. You can think of at least using the metrics proposed by MiDi (Vegnac et al23). But since this is pocket-conditional generation task, we are also interested in how to do the generated ligand bind to the pocket and how stable it is (eg metrics related to docking score, strain energies, protein-ligand interactions, etc). See for example the metrics in TargetDiff and in PoseCheck).

**Questions:**

- See above for questions
- It would be nice to see this model applied in other diffusion-based generative model (eg SBDDDiff), since it should be agnostic to the type of diffusion model used. Did the authors try any of these experiments?

---

> ### Author Response · Authors · 2023-11-22
> **Rebuttal 1**
>
> Thank you for taking the time to read and review our paper! Please see our responses below. We have started working on the corresponding changes. I hope that our response has sufficiently addressed your inquiry. All our modifications have been distinctly marked in blue font within the main body of the text.
> **W1**: We sincerely appreciate the reviewer's comments and have made the necessary improvements to the revised manuscript accordingly.
>
> **W2**: Lack details: Thanks to the reviewer for the suggestions. In response to these valuable comments, we have taken the initiative to provide a more detailed description in our manuscript regarding the computational process. Notably, we have employed the LBFGS as the solver for the proximal objective function. Furthermore, for the convenience of readers and to provide a more comprehensive understanding, we have included the pseudocode in the appendix.
>
> **W3**: Cut-off 2Å: The reviewer might have some misunderstanding of PoseCheck (Harris et al23). Recalling the PoseCheck, They define a clash as occurring when the pairwise distance between a protein and ligand atom falls below the sum of their van der Waals radii, with a clash tolerance of 0.5Å. For instance, Taking into account that the C-C bond distance is 1.5Å, adding the 0.5Å threshold results in 2.0Å. Given that common chemical bonds are typically smaller than 2.0Å, we believe our 2Å cut-off is a reasoning choice, rather than an arbitrary one.
>
> **W4**: Calculate clash: We apologize if our earlier explanation concerning the calculation of clashes was unclear. Let us offer a more comprehensive clarification. We have established a threshold of 2.0 Å. As a result, if the pairwise distance between any protein atoms and molecule atoms falls below this threshold, we regard the molecule as having a clash. The simplified calculation formula is as follows: $d=min(norm(x_l -x_p))$, where $x_l$ and $x_p$ represent the positions of arbitrary atoms within the molecule and protein, respectively. A clash is detected when at least one pair of atomic distances between the molecule's and protein's atoms is less than the 2Å threshold. For each protein target, we generate 100 molecules and subsequently determine the clash rate by counting the frequency of clashes. In our revised manuscript, we have clarified the clash computation process in the evaluation section.
>
> **W5**: Evaluation metric: We are grateful for your insights. For both unconditional (de novo design) and conditional (Structure-Based Drug Design, SBDD) generation tasks, it is essential to ensure that the generated molecules possess optimal validity and connectivity. These metrics are crucial because they influence the applicability of the generated molecules in downstream tasks. Upon reviewing MiDi, we noticed that, besides atom stability, all other evaluation metrics have been employed in our experiments. Additionally, atom stability can be encompassed within connectivity. We have also adhered to the vina score metric mentioned in the Targetdiff model. In response to the reviewer's query about the docking score, docking score is a broad concept, and different algorithms will give out different docking scores. Here, we respectfully propose that, in our paper, we use Qvina2 (Alhossary et al., 2015) to measure the docking score and represent it as vina score. Since the protein-ligand interaction is a state determined by docking score and strain energy, it is regrettable that PoseCheck does not provide information on calculating strain energy. As a result, we cannot adopt their approach in this aspect.
>
> **Q1**: Other model：We utilize our method in the DiffSBDD model, and the outcomes (as shown in the table below) regarding the mean clash rate indicate the effectiveness of our approach in reducing the clash ratio across various models.
> | Time | Influence | Validity | Diversity | Novelty | Uniqueness | Mean CR |
> |:------:|:------------:|:----------:|:-----------:|:----------:|:----------------:|:-------------:|
> |         | normal     |	100% |  0.8246   | 100%	  |  97.44%	| 17.17%    |
> | 300   | K=1, b=10 |	99.78%  |	0.8277 | 100%	 | 96.82%	| 9.64% |
> | 200   | K=1, b=10 |100%     |	0.8284 |100%	 | 96.88%	 | 9.95% |
> | 150   | K=1, b=10 |100%     |	0.8263 |100%	 |  96.76% |10.37% |

---

> ### Comment · Reviewer_xcbL · 2023-11-22
>
> We thank the authors for their feedback. We appreciate the extra clarifications and we would recommend the authors to add them on a future revision of the manuscript. I still disagree with the choice of metrics used to benchmark methods. In particular, the metrics used by the authors are not similar to MiDi (many of the more interesting metrics related to the distances of distributions are missing).
> For the reasons above, I keep my rating as is.

---

> > ### Author Response · Authors · 2023-11-23
> > **Rebuttal2**
> >
> > We thank your feedback!
> >
> > In our related work section, we have discussed the MiDi. The reviewers have pointed out certain metrics that, in our opinion, are challenging to apply to tasks focused on structure-based generation. To begin with, these metrics require a comparison against a specific test set. However, in the case of the strucuture-based drug design (SBDD) task, the test set comprises only pocket data, making it impractical to compare generated molecules directly with the pocket. Furthermore, the primary objective of the SBDD task is to identify novel molecular targets for proteins, rather than ensuring that the generated molecules closely resemble the original ligands.

---

### Meta-Review · Area_Chair_DHM9 · 2023-12-13

**Metareview:**

The paper proposes a scheme for reducing atomic clashes in solutions produced by diffusion models for Structure-Based Drug Design by including physical constraints. The reviewers felt that the method introduced was intuitive and clear but there was some concern regarding limited novelty. There was greater concern regarding the thoroughness of empirical evaluation of the method, and especially the choice of evaluation metrics. Three of four reviewers recommend rejection.

**Justification For Why Not Higher Score:**

The authors made a significant attempt to address reviewer criticism, however concerns regarding the degree of biological relevance of the evaluation metrics remained, casting doubt on the significance of the improvements resulting from the proposed method.

**Justification For Why Not Lower Score:**

NA

---

### Decision · Program_Chairs · 2024-01-16

Reject